# Safety and Mobility Performance Comparison of Two-Plus-One and Two-Lane Two-Way Roads: A Simulation Study

**Usama Elrawy Shahdah [1], Fayez Alanazi [2,*], Abdelhalim Azam [2,*] and Marwa Elbany [3]**

[1] Public Works Engineering Department, Faculty of Engineering, Mansoura University, Mansoura 35116, Egypt; usama.shahdah@mans.edu.eg
[2] Civil Engineering Department, College of Engineering, Jouf University, Sakaka 72388, Saudi Arabia
[3] Public Works Engineering Department, Faculty of Engineering, Port Said University, Port Said 42526, Egypt; mr_elbany@eng.psu.edu.eg
[*] Correspondence: fkalanazi@ju.edu.sa (F.A.); amazam@ju.edu.sa or abdelhalim.azam@mans.edu.eg (A.A.); Tel.: +966-546788031 (A.A.)

**Abstract:** Two-plus-one (2+1) highways are a special configuration of two-lane two-way (TLTW) highways with a continuous center lane that is used to alternate passing lanes. The main objective of this paper is, therefore, to evaluate the suitability of the 2+1 design for Middle East conditions as a replacement for traditional TLTW roads with higher traffic volumes or as an interim solution before transforming TLTW roads into four-lane highways. In our analysis, we considered both safety and mobility performances by comparing the 2+1 and TLTW designs. The new suggested 2+1 designs were evaluated, with the first design prohibiting overtaking in the opposite direction, while the second design permitted it. Additionally, two-speed-limit strategies, uniform speed limit (USL), and differential speed limit (DSL) were also evaluated. The results showed that the 2+1 design, which prohibited overtaking in the opposite direction, was superior to TLTW in terms of mobility and safety, while the other design compromised safety compared to TLTW. The results provide valuable information to policymakers, urban planners, and transport authorities to guide evidence-based decisions on the integration of the 2+1 design as a viable solution for sustainable and efficient transportation.

**Keywords:** 2+1 highways; two-lane roads; road safety; SUMO; surrogate safety measures; time to collision

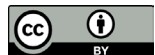

## 1. Introduction

A two-lane two-way (TLTW) road is an undivided road with one lane in each direction [1]. TLTW roads are prevalent throughout the transportation network, serving as vital arteries connecting urban centers, rural communities, and key economic hubs. While TLTW roads provide essential connectivity, they also present challenges, particularly in areas with high traffic volumes or limited infrastructure. Congestion and safety concerns are common, as the absence of a physical barrier between lanes can lead to head-on collisions and risky overtaking maneuvers. Efforts to improve safety on these roads include enhanced signage, road markings, and traffic enforcement measures aimed at mitigating risks and promoting responsible driving behavior. Investment in infrastructure upgrades and safety initiatives is essential to maximize the benefits of TLTW roads and minimize their associated challenges.

Transportation policies have historically focused on infrastructure development to improve road networks and enhance public transportation systems [2]. However, chal-

lenges persist due to rapid urbanization, inadequate enforcement of traffic laws, and limited resources for maintenance and expansion that should be investigated. Driver behaviors often reflect a mix of cultural norms, economic pressures, and enforcement gaps that result in issues such as aggressive driving, disregard for traffic signals, and high rates of traffic accidents [3]. Additionally, roadways vary widely in quality, with major highways often well-maintained, but urban streets and rural roads face challenges, such as congestion, poor signage, and lack of pedestrian infrastructure. Addressing these issues requires a holistic approach that combines policy interventions, enforcement measures, public awareness campaigns, and investment in infrastructure upgrades. One such upgrade is using 2+1 lane highways, which needs more studies to investigate the significance of mobility and safety on the existing roadway performance.

Two-plus-one (2+1) highways are also known as three-lane highways that aim to improve road safety and traffic flow on existing TLTW highways. This concept involves the addition of a middle lane (i.e., auxiliary lane) for alternating use in both directions, allowing for safer overtaking opportunities. The passing lane (i.e., auxiliary lane) is the main feature of the 2+1 designs. It allows drivers to comfortably overtake other vehicles [4], and it can increase travel speed and reduce risky overtaking maneuvers [5]. A report, by The National Cooperative Highway Research Program (NCHRP) (2003), reviewed the safety and operational experience of Germany, Finland, and Sweden with a 2+1 design and found that this design is most likely to be effective in terms of both safety and mobility and can be a good alternative to TLTW or four-lane roads in some cases [6].

The design of 2+1 roads can allow for safer overtaking opportunities, reducing the likelihood of head-on collisions common on single-lane roads. These roads are particularly beneficial in areas with a heavy traffic volume or challenging terrain, providing a balance between capacity and safety.

The main design features of the 2+1 include (1) the median separation type (in case of prohibiting overtaking in the opposite direction), (2) the length of the passing auxiliary lane, and (3) the width of the auxiliary lane. Calvi et al. (2023), in their driving simulator study, tested four median types: (1) double-line markings only; (2) reflective elements; (3) flexible guideposts; and (4) cable barriers and found that the type of median significantly affects driving behavior, but did not affect the average speed on the passing lane [7]. In a similar study, Vadeby (2016) investigated the safety effects of narrow 2+1 roads (i.e., a width of 9-10 meters) with cable barriers, in Sweden, and found that the total number of fatalities and serious injury crashes was reduced by 50%, and the total number of personal injury crashes decreased by 21% [8].

The length of the passing auxiliary lane is an important feature of the "2+1" design that would affect both safety and mobility [9] and is considered the main feature of the 2+1 designs, which allows drivers to comfortably overtake other vehicles [4]. A passing-lane length of less than 1000 meters is not recommended [9,10]. Internationally, the passing-lane length is between 800 to 3500 meters [11]. Durth (1995) recommended that passing-lane section lengths between 1000 and 1400 meters for volumes of up to 1000 vehicles per hour per direction, with heavy vehicles (HVs) percentages of up to 15% [12]. Furthermore, the Swedish design guidelines recommend that overtaking sections be 1–2.50 km long [13]. Lastly, the range of lane widths used internationally for a single passing lane within the 2+1 design was between 3.25 and 3.75 meters [11].

Harwood et al. (1988) provided instructions for identifying, constructing, and marking passing lanes efficiently to enhance traffic operations [14]. In addition, they provided, with other researchers, a method for determining passing lanes' operational efficiency in terms of enhanced service [14,15].

Kieć (2017) simulated the effect of heavy vehicles (HVs) shares (up to 20%) in the traffic stream of a 2+1 long road bypass, using PTV VISSIM [16], and found that the 2+1 system appears to be reasonably effective when compared to the TLTW bypass for traffic volumes up to an average annual daily traffic (AADT) of 22,000 [17]. Furthermore, Kirby

et al. (2014) recommended that speed limits for the 2+1 design be between 80 and 100 km/h [9].

Moreover, internationally, some 2+1 designs prohibit overtaking in the opposite direction, while others permit it [11]. One of the main benefits of the 2+1 design, which prohibits overtaking in the opposite direction, is the elimination of head-on collisions, which are the main threat in the conventional TLTW design. In Poland, the introduction of a 2+1 design instead of a conventional two-lane road design for treated sites that have 16 segments and extends 12.9 km long could reduce total crashes by 45% [18]. In addition, Harwood and John (1985) found that using passing lanes could potentially reduce the accident rate by about 9% and 17% for total and fatal and injury crashes, respectively, based on data from 22 passing sites from different states in the USA [19]. They reported that these reductions were not statistically significant, as there was only one year of accident data in the after period [19]. The Swedish experience of installing a 2+1 fence with cable barriers along the median was examined by Larsson et al. (2003) [20], who reported reductions of up to 50% for all crash types and up to 90% for fatal crashes [20]. According to the NCHRP (2003) report, which focuses on the application of the European 2+1 roadway designs, there was a 25% decrease in all crash types in Finland, with a 45% decrease in fatal crashes [6].

As far as the authors are aware, there are presently no Middle Eastern countries that have single 2+1 roads. Hence, the primary objective of this study is to evaluate the 2+1 roads for Middle Eastern conditions and assess their safety and mobility performance compared to the conventional TLTW roads. Different design scenarios were considered: (1) a design that allows opposite-direction overtaking and (2) another design that prohibits opposite-direction overtaking. Moreover, different speed-limit strategies were investigated, namely, (1) the uniform speed-limit (USL) strategy, in which all vehicles (i.e., passenger cars and heavy vehicles) operate with the same speed limit and (2) the differential speed-limit (DSL) strategy, where large vehicles and other types of vehicles operate with different speed limits. The DSL strategy is the chosen strategy for speed limits on Eastern countries' highways. For example, on Egyptian highways, the truck trailers' speed limit is restricted to 30 km/h lower than the speed limit for passenger cars [21]. In the current analysis, the mobility efficiency was measured in terms of average travel speed (km/h) and average delay (seconds/vehicle), while traffic safety was assessed using simulated conflicts. The time-to-collision (TTC) values ≤ 2.50 seconds for low-severity conflicts, TTC ≤ 1.50 seconds for moderate conflicts, and TTC ≤ 0.50 seconds for severe conflicts were generated using the SUMO model.

## 2. Simulation Methodology

The safety of transportation networks can be assessed using surrogate safety metrics that come from microscopic traffic simulations (e.g., [21–29], etc.). Using simulation modeling has many benefits, one of which is the ability to test potential treatments prior to implementation [30]. Although, most micro-simulation software, such as VISSIM [16], PARAMICS [31], AIMSUN [32], Simulation Of Urban Mobility (SUMO) [33,34], INTEGRATION [35], CORSIM [36], MITSIMLab [37], etc., can be used to model different types of roads and 2+1 designs [38,39], almost all of them cannot model driving (i.e., overtaking) in the opposite direction.

The SUMO microscopic traffic simulation model [33,34] can model overtaking in the opposite direction. In addition, it provides all the modeling features required to put in place suitable substitute safety precautions. Therefore, in this study, the evaluation of the safety and mobility effects of the 2+1 and the conventional TLTW roads were assessed using the SUMO model. For vehicles equipped with Surrogate Safety Measure (SSM) devices, SUMO enables the extraction of simulated conflicts as a direct output of the simulation. SUMO can also produce floating car data (FCD), which can be used by the Surrogate Safety Assessment Model (SSAM) software to obtain conflicts [40].

Several substitute safety indicators can be used when conducting road safety studies using micro-simulations, like time-to-collision (TTC), deceleration rate to avoid the collision (DRAC) [41], time-to-accident (TTA) [42], encroachment time (ET) [43], crash potential index (CPI) [23,30], etc., must be used [42,44]. In this analysis, the TTC is used as the surrogate safety indicator to reflect the crash potential, as it is simple to calculate. TTC can be defined as the time difference between two vehicles, assuming they follow their respective trajectories at their current speeds before they crash [44]. According to the lowest TTC value, the severity of a two-vehicle crash can be determined. Van der Horst (1990) [45] stated that, when the TTC value is less than 1.50 seconds, there may be a risk of a crash. Archer (2005) [46] also suggested TTC ≤ 1.50 s as a critical value for road safety in urban areas. The AASHTO (2011 and 2018) suggested that the TTC value for brake reaction time, which is utilized in the construction of highway stopping sight distances, be 2.5 s [47,48].

## 3. Study Framework

The SUMO [33] microscopic traffic simulation model was used to obtain and model the vehicle trajectories, simulated speeds, simulated delays, and simulated traffic conflicts. In this analysis, the most popular car-following model in the literature is the Wiedemann-99 model (e.g., [21,23,25,49–53], etc.), which was adapted to this analysis. A full 90 min, or 5400 s, of traffic were simulated. Ten simulation runs, each using a different random seed, were applied for each scenario, with a 15 min warm up. Only the simulation results of the middle 60 min are used, after removing the 15 min warming up and the final 15 min.

The road network with traffic and geometric characteristics can be used to determine the average delay and traffic speed after the simulation has run, and each simulation step can yield the trajectories of every vehicle. The number of simulated conflicts is then obtained by transforming the individual vehicle trajectories into pairs of vehicles for a specific interaction type (i.e., leading and trailing vehicles for rear-end interaction and head-on interactions). Additionally, car locations with velocity and acceleration profiles were obtained every 0.50 s, and an action-step length of 0.50 s was used. Vehicle lengths of 5.00 m and 18.00 m were used for both passenger cars (PCs) and heavy vehicles (HVs) or tractor–trailers (TTs), respectively.

Figure 1 illustrates a general framework for calculating traffic delays and conflicts. The first step involves simulating all vehicle movements (PCs and HVs) on the highway network using SUMO. The TLTW or 2+1 geometry, number of lanes, and lane configuration represent the input data. The combinations of traffic volumes and the percentage of HVs with the speed-limit strategy (i.e., USL or DSL) are displayed in Table 1.

**Table 1.** Different combinations of traffic volumes, %HVs, and speed limits.

| Speed-Limit Strategy | Traffic Volume (Vehicle/Hour/Direction) | HVs (% from Total Volume) | Speed Limit |
|---|---|---|---|
| USL | 250<br>500<br>750<br>1000<br>1250 | 2.50, 5.0, 10.0 and 15.0 | - 50 km/h<br>- 60 km/h<br>- 70 km/h<br>- 80 km/h<br>- 90 km/h |
| DSL | 250<br>500<br>750<br>1000<br>1250 | 2.50, 5.0, 10.0 and 15.0 | PCs: 90 km/h<br>- HVs: 60 km/h<br>PCs: 80 km/h<br>- HVs: 60 km/h |

**Highway Geometry:**
- Number of lanes
- Lane configuration
- Lane width
- Location of additional lanes

**Traffic Characteristics:**
- Traffic volume in both directions
- % Heavy Vehicles (HVs)
- % Passenger Cars (PCs)
- Speed limit for PCs & HVs

**Drivers' behaviors parameters:**
- Car following
- Gap acceptance
- Lane-change

**Traffic Micro-Simulation Model (SUMO)**

- **Simulated Delay**
- **Simulated Travel Time**
- **Simulated Speed**

**Pair-Vehicles' Trajectories**

**Extractions of vehicles' Interactions (SSM Devices)**

Vehicles' Interactions

High-Risk Interactions

Low-Risk Interactions

Undisturbed Passages (i.e., no interactions)

**Conflicts Thresholds:**
-TTC≤2.50s
-TTC≤1.50s
-TTC≤0.50s

**Surrogate Safety Measure**
- Time-to-Collision (TTC)

**Simulated Traffic Conflicts**

**Figure 1.** Study framework.

Figure 2 shows the geometry of the highway networks, with the following assumptions: (a) lane width is 3.65 m and (b) alternating lane with 1.0 km length for both 2+1 designs.

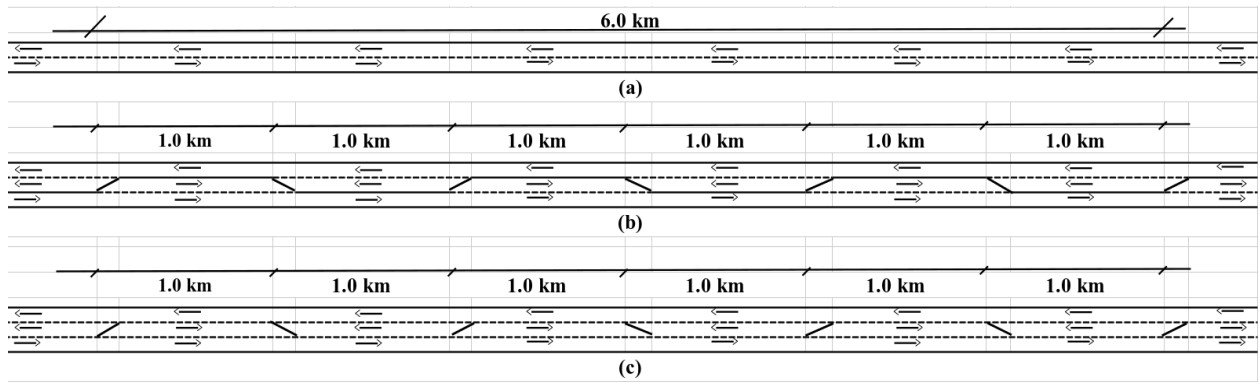

**Figure 2.** Traffic network: (**a**) convectional TLTW, (**b**) 2+1 (prohibited overtaking in opposite direction), and (**c**) 2+1 (allowed overtaking in opposite direction).

### 3.1. Traffic Conflict Ratio:

To evaluate the 2+1 designs against the conventional TLTW design for different traffic conditions, the simulated conflicts in the before period (i.e., with the conventional TLTW design) and the after period (i.e., with the 2+1 design) are estimated, and then the conflict ratio ($\rho$) is obtained as follows [26,27]:

$$\rho = \frac{C_a}{C_b} \tag{1}$$

with a variance of:

$$Var(\rho) = (C_a/C_b)^2 \times \left[ \left( Var\,(C_a)/C_a{}^2 \right) + \left( Var\,(C_b)/C_b{}^2 \right) \right] \tag{2}$$

where,
$\rho$ = Conflict ratio;
$C_a$ = Average number of conflicts for the 2+1 design (i.e., after);
$C_b$ = Average number of conflicts for the conventional TLTW design (i.e., before);
$Var(C_a)$ = Variance of the conflicts in the after;
$Var(C_b)$ = Variance of the conflicts in the before.

The "$\rho$" value can be viewed as a modification factor based on conflicts [28]. When the value of " $\rho$ " is higher than "1.0", it indicates that the 2+1 design would result in more simulated conflicts and, hence, more accidents than the TLTW design. On the contrary, when the value of " $\rho$ " is lower than "1.0", it indicates that the "2+1" design would result in fewer simulated conflicts. Consequently, fewer accidents would occur compared to the TLTW design. A " $\rho$ " value of "1.0" indicates no significant safety changes between the two designs. The statistical significance of the " $\rho$ " can be estimated in the same fashion as the collision modification factors (CMFs) [21,26,27,54]. For the 5% significance level, it can be estimated as follows) [55,56]:

$$CI_{95\%} = \rho \pm 1.96 \times SE \tag{3}$$

where
$CI_{95\%}$= the confidence interval at the 5% significance level;
$SE$ = the standard error from the 10 simulations.

### 3.2. SUMO Calibration

A one-hour video was recorded for a TLTW road section of 375 m from the Mansoura–Damietta Road in Egypt between 12:00 and 1:00 p.m. on 27 June 2021. The total road width is 8.50 m, with 1.50 m and 1.00 m widths for the right and the left shoulders, respectively. Table 2 shows a summary of the traffic characteristics of the road section during the one-hour period.

**Table 2.** Summary of the observed traffic characteristics.

| Parameter | 1st Direction | 2nd Direction |
|---|---|---|
| Traffic volume (vehicle/h) | 189 | 212 |
| Average travel speed ± standard deviation | 64.39 ± 11.06 km/h 17.88 ± 03.07 m/s | 66.42 ± 13.94 km/h 18.45 ± 03.87 m/s |
| Number of successful overtakes | | 32 |

In this analysis, for the SUMO simulation, the Wiedemann-99 car-following and the LC2013 lane-change models were adapted. The tested values for both the car-following and the lane-change parameters are shown in Table 3. A Python script was developed with several loops for each parameter range. A single simulation run for each combination of parameters was executed. Then, the combination that has the lowest difference between the simulated travel speed and the observed travel speed in both directions, and at the same time has the lowest difference between the simulated number of overtakes and the

observed number of overtakes, was chosen. Then, 10 simulation runs for the selected combination were executed to make sure that the calibrated parameters can successfully represent the range of observed speed and observed number of overtakes. The calibrated car-following and lane-change parameters are shown in Table 3.

**Table 3.** Simulation Parameters.

| Parameter | Description | Default Values | Tested Values | Calibrated Values |
|---|---|---|---|---|
| Car-Following Parameters (Wiedemann-99) | | | | |
| CC0 | Standstill gap (m) | 2.50 | 1.50 * | 1.50 |
| CC1 | Headway time: Time gap that the following driver keeps for a safety-in-moving state (s) | 1.30 | 0.50–1.50 (step = 0.1) | 1.00 |
| CC2 | 'Following' variation: Range of gap between the two vehicles in the "following" regime (m) | 8.00 | 4.0–10.0 (step =1.0) | 4.50 |
| CC3 | Threshold for entering "following" regime (s) | −12.00 | −6.00–−12.00 (step = 1.00) | −12.00 |
| CC4 | Negative 'following' threshold (m/s) | −0.25 | −0.25 * | −0.25 |
| CC5 | Positive 'following' threshold (m/s) | 0.35 | 0.35 * | 0.35 |
| CC6 | Speed dependency of oscillation ($10^{-4}$ rad/s) | 6.00 | 6.00–12.00 (step = 1.0) | 6.00 |
| CC7 | Oscillation acceleration: Actual acceleration during oscillation in the unconscious following regime (m/s²) | 0.25 | 0.25 * | 0.25 |
| CC8 | Standstill acceleration: Desired acceleration when the vehicle starts from the standing condition (m/s²) | 2.00 | 1.50–3.50 (step = 0.50) | 2.00 |
| CC9 | Desired acceleration at 80 km/h (m/s²) | 1.50 | 1.00–2.00 (step = 0.5) | 1.50 |
| Lane-Change Parameters (LC2013) | | | | |
| lcOpposite | The eagerness to overtake through the opposite-direction lane. Higher values result in more lane changing. | 1.00 | 1.00–10.00 (step = 1.0) | 1.00 |
| lcSpeedGain | The eagerness to perform lane changing to gain speed. Higher values result in more lane changing. | 1.00 | 1.00–10.00 (step = 1.0) | 1.00 |
| lcOvertakeDeltaSpeedFactor | Speed difference factor for the eagerness to overtake a neighbor vehicle before changing lanes. | 0.00 | −1.00–1.00 (step = 0.25) | 0.00 |
| lcStrategic | The eagerness for performing strategic lane changing. | 1.00 | 0.00–2.00 (step = 0.1) | 1.10 |
| lcSpeedGainLookahead | Lookahead time in seconds for anticipating slow down (s). | 0.00 | 0.00–10.00 (step = 1.00) | 0.00 |
| lcCooperative | The willingness to perform cooperative lane changing | 1.00 | 0–1.00 (step = 0.25) | 1.00 |

\* This parameter was fixed.

As shown in Table 4, the relative difference between the simulated and the observed traffic volume and traffic speed are within the 10% error, and the observed number of overtakes is within the range of the simulated ones, which suggests that the calibrated parameters would be good representatives of the TLTW road from which the data was collected.

**Table 4.** Summary of the simulated traffic characteristics.

| Parameter | 1st Direction | 2nd Direction |
|---|---|---|
| Traffic volume (vehicle/h) | 189.10 ± 0.57 (0.00%) | 212.20 ± 0.63 (0.00%) |
| Average travel speed ± standard deviation (meter/second) | 19.26 ± 0.31 (7.68%) | 19.23 ± 0.40 (4.24%) |
| Number of successful overtakes | 35.40 ± 3.53 (10.62%) | |

Values between parentheses represent the relative difference between the simulated and the observed values.

## 4. Study Results:

### 4.1. Uniform Speed-Limit (USL) Results

#### 4.1.1. Simulated Mobility Results

Figure 3 shows the average network delay (second/vehicle) for the three design alternatives (i.e., TLTW, 2+1 that prohibits overtaking in the opposite direction, and 2+1 that allows overtaking in the opposite direction) using the USL strategy with speed limits of 50, 60, 70, 80, and 90 km/h for different traffic volumes and different percentages of HVs. It is obvious that the percentage of HVs does not have an effect at all on the average delay. There are slight delay differences for traffic volumes of around 1250 vehicles per hour, but all the differences are not statistically significant at the five percent level of significance.

In addition, both of the 2+1 designs are expected to significantly reduce the average delay compared to the conventional TLTW design for all traffic volumes and all tested speed limits. Those changes are all statistically significant at the five percent level of significance. Furthermore, the 2+1 design that allows overtaking in the opposite directions results in a higher reduction in the average delay compared to the 2+1 design that prohibits overtaking in the opposite direction. The differences in delay between the 2+1 designs are negligible for higher speed limits and higher traffic volumes, as shown in Figure 3.

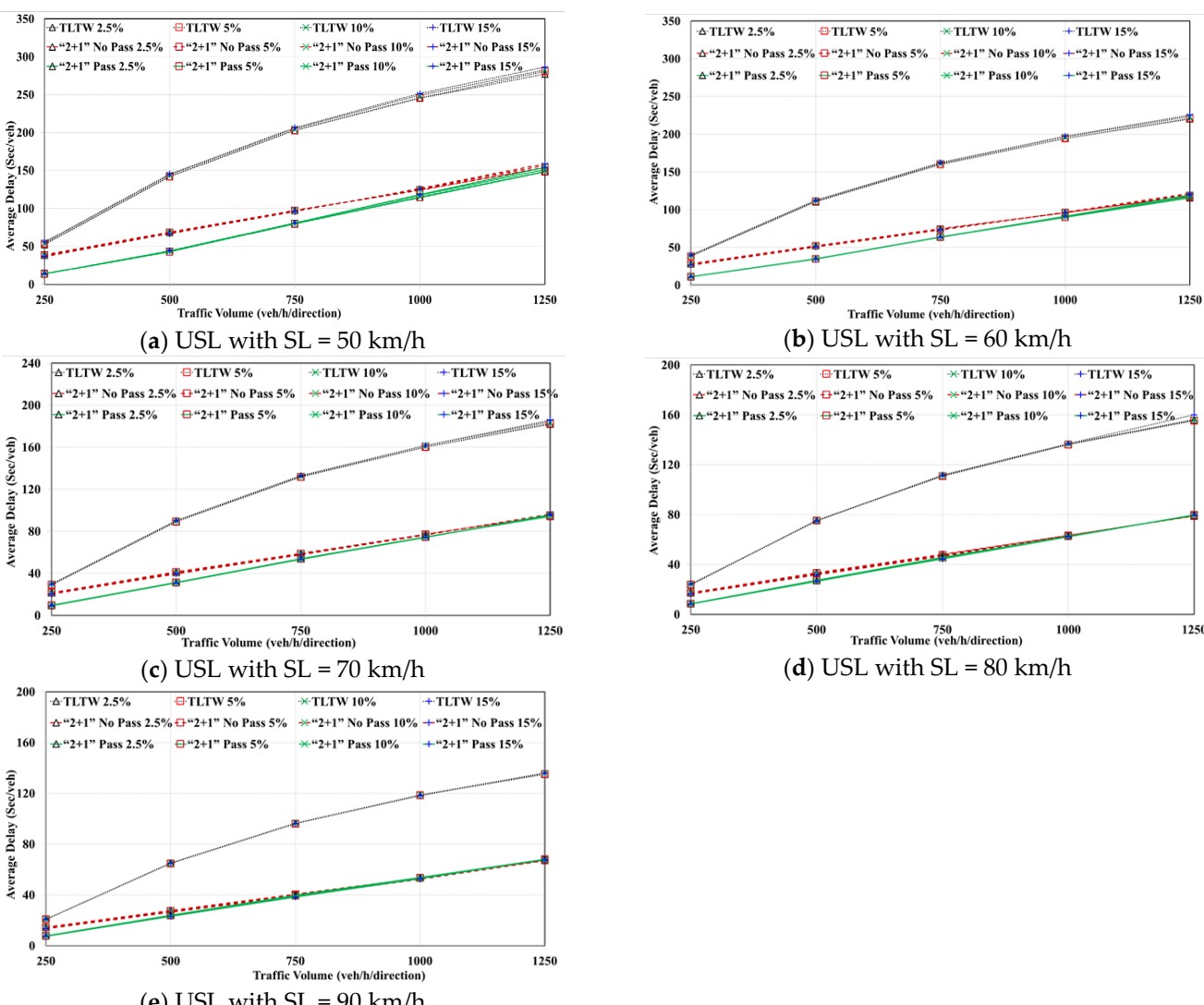

**Figure 3.** Average network delay (second/vehicle) for uniform speed-limit scenarios: (**a**) speed limit = 50 km/h, (**b**) speed limit = 60 km/h, (**c**) speed limit = 70 km/h, (**d**) speed limit = 80 km/h, and (**e**) speed limit = 90 km/h.

Similarly, Figure 4 shows the average travel speed (ATS) results for the USL strategy. It shows concise results that are similar to the average delay results, in that the %HVs has no effect and the ATS increases significantly for the two 2+1 designs compared to the conventional TLTW design, and there is a negligible difference in travel speed between the 2+1 designs for higher speed limits and higher traffic volumes.

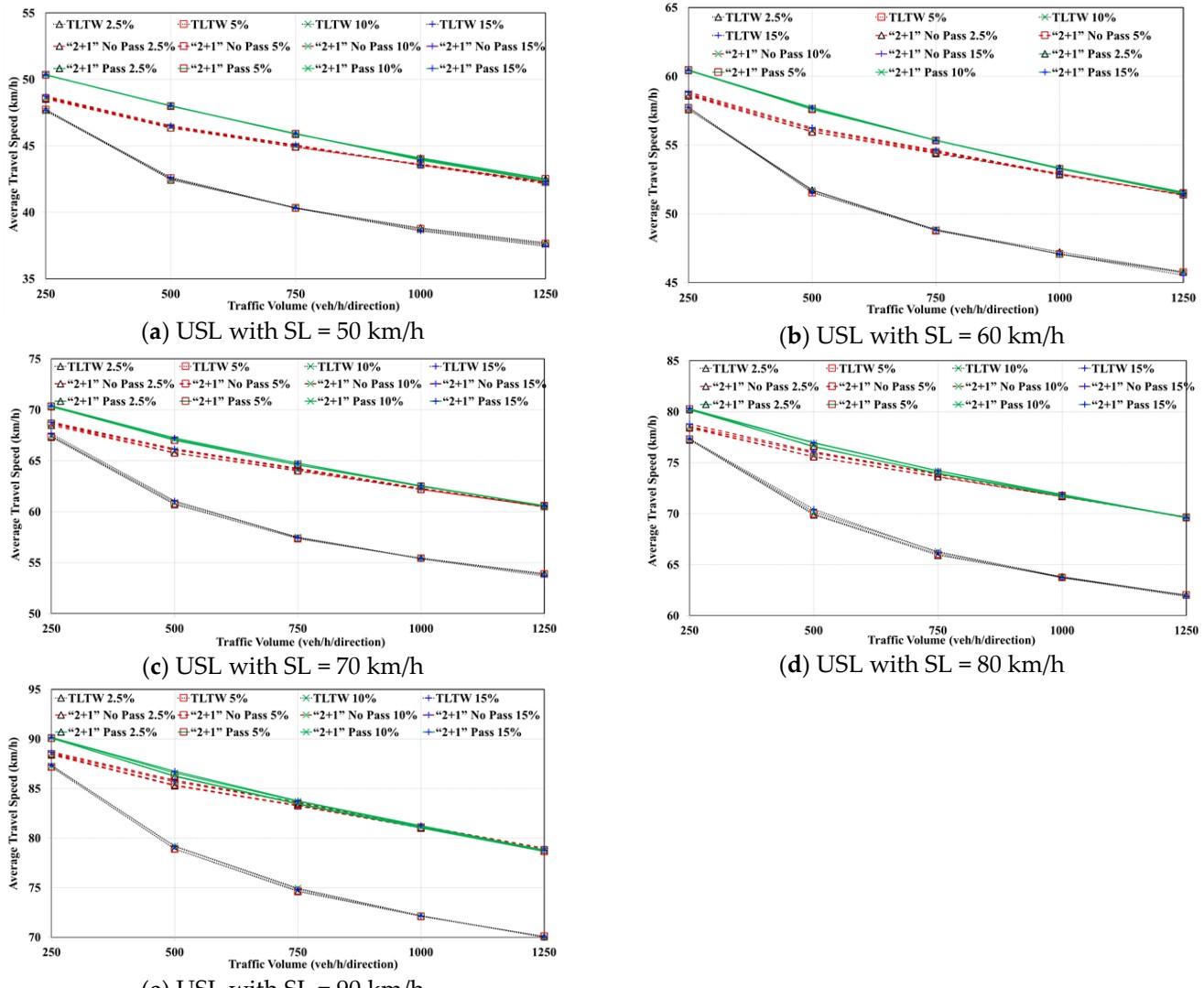

**Figure 4.** Average travel speed (km/h) for uniform speed-limit scenarios: (**a**) speed limit = 50 km/h, (**b**) speed limit = 60 km/h, (**c**) speed limit = 70 km/h, (**d**) speed limit = 80 km/h, and (**e**) speed limit = 90 km/h.

Table 5 shows the results of the USL strategy for different speed limits and different traffic volumes at 2.5% HVs. For both 2+1 designs, there are statistically significant improvements, at the 5% significance level, in both ATS (i.e., increase) and delay (i.e., decrease) for different traffic volumes and different speed limits compared to the TLTW design. The improvements in ATS were between 1.4% (at low traffic volume) and 12.5% (at volume near capacity) and between 3.3% and 13.80% for the 2+1 design that prohibits overtaking in the opposite direction and the one that allows it, respectively. In addition, the reduction in average delay varies from 26% to 50% and from 47% to 74% for the 2+1 design that prohibits overtaking and the one that allows it, respectively.

**Table 5.** The average travel speed (km/h) and the average delay (sec/veh), ± standard deviation, and the % relative difference with the TLTW design Results for USL strategy for 2.50% HVs.

| Speed Limit | V | TLTW | | 2+1 (No Pass) | | 2+1 (Allow Pass) | |
|---|---|---|---|---|---|---|---|
| | | Speed | Delay | Speed | Delay | Speed | Delay |
| 50 | 250 | 47.71 ± 0.26 | 52.40 ± 2.96 | 48.52 ± 0.24 (1.7) | 39.00 ± 2.23 (−26) | 50.34 ± 0.22 (5.5) | 13.65 ± 0.59 (−74) |
| | 500 | 42.44 ± 0.17 | 141.81 ± 5.54 | 46.36 ± 0.13 (9.2) | 68.68 ± 2.02 (−52) | 48.01 ± 0.11 (13.1) | 42.95 ± 1.16 (−70) |
| | 750 | 40.33 ± 0.18 | 202.85 ± 7.41 | 44.92 ± 0.12 (11.4) | 97.41 ± 2.54 (−52) | 45.87 ± 0.11 (13.8) | 79.61 ± 2.23 (−61) |
| | 1000 | 38.83 ± 0.12 | 245.19 ± 4.99 | 43.57 ± 0.10 (12.2) | 124.02 ± 2.66 (−49) | 44.08 ± 0.10 (13.5) | 114.28 ± 2.52 (−53) |
| | 1250 | 37.71 ± 0.10 | 277.07 ± 4.38 | 42.27 ± 0.08 (12.1) | 153.59 ± 2.36 (−45) | 42.51 ± 0.09 (12.7) | 148.19 ± 2.59 (−47) |
| 60 | 250 | 57.58 ± 0.24 | 38.37 ± 2.19 | 58.58 ± 0.25 (1.7) | 28.31 ± 1.34 (−26) | 60.42 ± 0.26 (4.9) | 11.08 ± 0.46 (−71) |
| | 500 | 51.73 ± 0.23 | 110.38 ± 5.38 | 55.96 ± 0.15 (8.2) | 52.20 ± 1.90 (−53) | 57.62 ± 0.15 (11.4) | 34.73 ± 1.16 (−69) |
| | 750 | 48.83 ± 0.18 | 159.98 ± 4.67 | 54.39 ± 0.16 (11.4) | 74.59 ± 2.47 (−53) | 55.36 ± 0.16 (13.4) | 63.47 ± 2.20 (−60) |
| | 1000 | 47.23 ± 0.15 | 194.36 ± 4.63 | 52.91 ± 0.14 (12.0) | 95.97 ± 2.24 (−51) | 53.34 ± 0.15 (12.9) | 89.83 ± 2.24 (−54) |
| | 1250 | 45.78 ± 0.12 | 220.21 ± 3.22 | 51.39 ± 0.11 (12.3) | 118.20 ± 1.79 (−46) | 51.59 ± 0.12 (12.7) | 115.69 ± 2.21 (−47) |
| 70 | 250 | 67.31 ± 0.29 | 29.04 ± 1.88 | 68.48 ± 0.29 (1.7) | 21.62 ± 1.15 (−26) | 70.31 ± 0.29 (4.4) | 9.42 ± 0.62 (−68) |
| | 500 | 60.71 ± 0.22 | 89.17 ± 3.90 | 65.75 ± 0.18 (8.3) | 41.33 ± 1.71 (−54) | 67.02 ± 0.18 (10.4) | 31.28 ± 1.39 (−65) |
| | 750 | 57.41 ± 0.23 | 131.74 ± 4.08 | 64.01 ± 0.19 (11.5) | 58.90 ± 2.00 (−55) | 64.58 ± 0.19 (12.5) | 53.85 ± 1.93 (−59) |
| | 1000 | 55.44 ± 0.18 | 160.21 ± 3.64 | 62.19 ± 0.18 (12.2) | 77.11 ± 2.04 (−52) | 62.53 ± 0.17 (12.8) | 74.48 ± 1.88 (−54) |
| | 1250 | 53.93 ± 0.15 | 182.01 ± 2.84 | 60.50 ± 0.14 (12.2) | 95.05 ± 1.76 (−48) | 60.59 ± 0.14 (12.3) | 94.27 ± 1.84 (−48) |
| 80 | 250 | 77.23 ± 0.31 | 23.62 ± 1.74 | 78.40 ± 0.31 (1.5) | 17.43 ± 1.07 (−26) | 80.20 ± 0.31 (3.8) | 8.60 ± 0.45 (−64) |
| | 500 | 69.88 ± 0.32 | 75.42 ± 4.10 | 75.58 ± 0.22 (8.1) | 33.44 ± 1.32 (−56) | 76.60 ± 0.22 (9.6) | 27.25 ± 1.10 (−64) |
| | 750 | 65.92 ± 0.26 | 110.98 ± 3.71 | 73.61 ± 0.21 (11.7) | 48.18 ± 1.61 (−57) | 73.94 ± 0.21 (12.2) | 45.57 ± 1.58 (−59) |
| | 1000 | 63.82 ± 0.23 | 136.18 ± 3.32 | 71.69 ± 0.20 (12.3) | 63.37 ± 1.73 (−53) | 71.71 ± 0.22 (12.4) | 62.64 ± 2.00 (−54) |
| | 1250 | 62.00 ± 0.21 | 155.86 ± 3.74 | 69.67 ± 0.17 (12.4) | 79.09 ± 1.52 (−49) | 69.61 ± 0.16 (12.3) | 79.37 ± 1.50 (−49) |
| 90 | 250 | 87.18 ± 0.33 | 20.80 ± 1.17 | 88.40 ± 0.32 (1.4) | 14.61 ± 0.92 (−30) | 90.09 ± 0.33 (3.3) | 7.70 ± 0.54 (−63) |
| | 500 | 78.91 ± 0.32 | 65.05 ± 3.04 | 85.28 ± 0.23 (8.1) | 27.80 ± 1.06 (−57) | 86.23 ± 0.24 (9.3) | 24.01 ± 1.05 (−63) |
| | 750 | 74.61 ± 0.30 | 96.45 ± 3.18 | 83.25 ± 0.25 (11.6) | 40.56 ± 1.62 (−58) | 83.42 ± 0.28 (11.8) | 39.81 ± 1.68 (−59) |
| | 1000 | 72.10 ± 0.25 | 118.48 ± 2.76 | 81.01 ± 0.23 (12.4) | 53.42 ± 1.60 (−55) | 81.01 ± 0.24 (12.4) | 53.67 ± 1.69 (−55) |
| | 1250 | 70.05 ± 0.19 | 135.06 ± 2.08 | 78.81 ± 0.19 (12.5) | 67.17 ± 1.40 (−50) | 78.66 ± 0.18 (12.3) | 68.03 ± 1.29 (−50) |

4.1.2. Simulated Safety Results

Tables 6–10 show the conflict ratio ($\rho$) for the total simulated conflicts for time to collisions (TTC) of ≤2.5, ≤1.50, and ≤0.50 s and for speed limits of 50, 60, 70, 80, and 90 km/h, respectively, for the USL strategy. It is worth noting that, if $\rho > 1.0$, then the 2+1 design would increase the number of simulated conflicts and, hence, the number of crashes compared to the TLTW design, while a value of $\rho < 1.0$ indicates that the 2+1 design would reduce the number of simulated conflicts and, therefore, reduce the number of crashes compared to the TLTW design. And if $\rho = 1.0$, then the 2+1 design is not different from the TLTW design, and hence, there is no safety gain or loss.

It is worth noting that the total conflicts in the current analysis were selected, as most of conflicts in the TLTW design are head-on conflicts (i.e., vehicles would collide with each other's front bumper to front bumper), while in the 2+1 design with prohibited overtaking in the opposite direction, all conflicts are rear-end (the front bumper of the rear vehicle could collide with the rear bumper of the lead vehicle), which are different types of conflicts.

For the 2+1 design that prohibits overtaking in the opposite direction, the results (please refer to Tables 6–10) of the safety evaluation (i.e., the $\rho$ value compared to 1.0) show that the $\rho$ value for almost all scenarios (with different speed limits and traffic volumes) for different TTC thresholds is less than 1.0, which indicates a reduction in the simulated conflicts and, hence, a reduction in crashes and safety improvements. There are only a few values for TTC ≤ 2.50 s (i.e., low-severity conflicts) that show statistically significant values of $\rho > 1.0$ at the 5% level for higher traffic volumes (near capacity), which indicates an increase in the simulated conflicts when the TTC threshold is less than 2.50 s. These instances should not be viewed as safety deterioration, as drivers have 2.50 s to take an evasive action to avoid potential crashes, which should be enough for most drivers.

For the 2+1 with allowed overtaking in the opposite direction, the results show that the $\rho$ values, for almost all scenarios with different TTC thresholds, are statistically significantly greater than 1.0, which indicates an increase in the simulated conflicts and, hence, an increase in crashes.

**Table 6.** Conflict ratio, with standard error between conventional TLTW before and after 2+1 design for network speed of 50 km/h for USL Strategy.

| V | %HV | TLTW to 2+1 (No Pass) | | | TLTW to 2+1 (Allow Pass) | | |
|---|---|---|---|---|---|---|---|
| | | TTC ≤ 2.50 | TTC ≤ 1.50 | TTC ≤ 0.50 | TTC ≤ 2.50 | TTC ≤ 1.50 | TTC ≤ 0.50 |
| 250 | | 0.06 (0.04) | 0.03 (0.02) | 0.00 (0.00) | *1.98 (0.42)* | *2.75 (0.63)* | *2.41 (0.65)* |
| 500 | | 0.30 (0.05) | 0.03 (0.01) | 0.00 (0.00) | *2.52 (0.32)* | *5.43 (1.01)* | *6.13 (1.36)* |
| 750 | 2.5 | 0.56 (0.06) | 0.12 (0.03) | 0.03 (0.01) | *1.61 (0.15)* | *3.41 (0.41)* | *4.02 (0.50)* |
| 1000 | | **0.99 (0.11)** | 0.24 (0.04) | 0.03 (0.01) | *1.62 (0.17)* | *1.96 (0.27)* | *2.04 (0.29)* |
| 1250 | | 1.37 (0.08) | 0.60 (0.08) | 0.05 (0.02) | *1.67 (0.09)* | *1.86 (0.20)* | *1.83 (0.21)* |
| 250 | | 0.05 (0.03) | 0.03 (0.02) | 0.00 (0.00) | *2.09 (0.36)* | *2.79 (0.65)* | *2.30 (0.56)* |
| 500 | | 0.28 (0.05) | 0.05 (0.02) | 0.02 (0.02) | *2.83 (0.28)* | *5.29 (0.74)* | *6.78 (1.18)* |
| 750 | 5 | 0.63 (0.05) | 0.12 (0.04) | 0.04 (0.03) | *1.87 (0.11)* | *2.92 (0.33)* | *3.47 (0.41)* |
| 1000 | | **1.15 (0.11)** | 0.33 (0.07) | 0.03 (0.01) | *1.93 (0.16)* | *2.72 (0.42)* | *3.04 (0.55)* |
| 1250 | | *1.40 (0.13)* | 0.69 (0.13) | 0.07 (0.02) | *1.70 (0.16)* | *2.03 (0.37)* | *2.06 (0.43)* |
| 250 | | 0.06 (0.03) | 0.02 (0.02) | 0.00 (0.00) | *2.06 (0.44)* | *3.11 (0.92)* | *2.94 (1.04)* |
| 500 | | 0.27 (0.05) | 0.04 (0.02) | 0.02 (0.01) | *2.34 (0.40)* | *4.34 (1.05)* | *5.06 (1.46)* |
| 750 | 10 | 0.60 (0.08) | 0.10 (0.02) | 0.02 (0.01) | *1.73 (0.23)* | *2.81 (0.49)* | *3.14 (0.61)* |
| 1000 | | *1.14 (0.05)* | 0.33 (0.04) | 0.04 (0.01) | *1.75 (0.07)* | *1.99 (0.18)* | *2.18 (0.24)* |
| 1250 | | *1.64 (0.11)* | **0.91 (0.13)** | 0.08 (0.02) | *1.84 (0.12)* | *2.06 (0.29)* | *1.55 (0.29)* |
| 250 | | 0.05 (0.02) | 0.00 (0.00) | 0.00 (0.00) | *1.97 (0.25)* | *3.26 (0.73)* | *3.20 (0.86)* |
| 500 | | 0.35 (0.06) | 0.04 (0.02) | 0.01 (0.01) | *2.53 (0.39)* | *4.52 (0.93)* | *4.92 (1.14)* |
| 750 | 15 | 0.76 (0.08) | 0.12 (0.04) | 0.01 (0.01) | *1.85 (0.15)* | *3.20 (0.38)* | *3.78 (0.62)* |
| 1000 | | **1.12 (0.09)** | 0.37 (0.07) | 0.04 (0.01) | *1.71 (0.14)* | *2.11 (0.31)* | *2.25 (0.33)* |
| 1250 | | *1.52 (0.10)* | **0.89 (0.11)** | 0.09 (0.02) | *1.73 (0.11)* | *1.80 (0.20)* | *1.52 (0.20)* |

*Bold* indicates that $\rho$ values are not statistically significantly different than one (i.e., no safety changes between the 2+1 and the TLTW designs) at the 5% significance level. *Italic* (Red) indicates that $\rho$ values are statistically significant >1.0 (i.e., reduced safety of the 2+1 design compared to the TLTW design) at the 5% significance level. *Neither Bold nor Italic* indicates that $\rho$ values are statistically significant <1.0 (i.e., improved safety of the 2+1 design compared to the TLTW design) at the 5% significance level.

**Table 7.** Conflict ratio, with standard error between conventional TLTW before and after 2+1 design for network speed of 60 km/h for USL Strategy.

| V | %HV | TLTW to 2+1 (No Pass) | | | TLTW to 2+1 (Allow Pass) | | |
|---|---|---|---|---|---|---|---|
| | | TTC ≤ 2.50 | TTC ≤ 1.50 | TTC ≤ 0.50 | TTC ≤ 2.50 | TTC ≤ 1.50 | TTC ≤ 0.50 |
| 250 | | 0.19 (0.07) | 0.02 (0.02) | 0.00 (0.00) | *2.76 (0.64)* | *2.94 (0.69)* | *3.23 (0.92)* |
| 500 | | 0.37 (0.05) | 0.04 (0.02) | 0.02 (0.01) | *2.93 (0.37)* | *3.61 (0.48)* | *3.99 (0.67)* |
| 750 | 2.5 | 0.72 (0.09) | 0.11 (0.02) | 0.02 (0.01) | *2.32 (0.26)* | *2.57 (0.27)* | *2.83 (0.38)* |
| 1000 | | 1.37 (0.15) | 0.26 (0.04) | 0.07 (0.02) | *2.15 (0.25)* | *1.63 (0.23)* | *1.70 (0.26)* |
| 1250 | | 1.70 (0.11) | 0.39 (0.04) | 0.07 (0.01) | *2.05 (0.13)* | *1.35 (0.13)* | **1.18 (0.14)** |
| 250 | | 0.27 (0.11) | 0.00 (0.00) | 0.00 (0.00) | *3.95 (1.16)* | *4.39 (1.33)* | *5.38 (1.85)* |
| 500 | | 0.34 (0.05) | 0.07 (0.02) | 0.02 (0.01) | *2.54 (0.31)* | *3.19 (0.43)* | *3.61 (0.45)* |
| 750 | 5 | 0.59 (0.06) | 0.09 (0.02) | 0.03 (0.01) | *1.71 (0.17)* | *1.85 (0.20)* | *2.01 (0.26)* |
| 1000 | | **1.19 (0.11)** | 0.21 (0.03) | 0.06 (0.01) | *1.88 (0.17)* | *1.44 (0.20)* | *1.54 (0.26)* |
| 1250 | | *2.15 (0.14)* | 0.47 (0.05) | 0.09 (0.02) | *2.45 (0.17)* | *1.25 (0.13)* | *1.05 (0.14)* |
| 250 | | 0.29 (0.07) | 0.00 (0.00) | 0.00 (0.00) | *3.33 (0.62)* | *3.58 (0.60)* | *4.19 (1.05)* |
| 500 | 10 | 0.38 (0.07) | 0.05 (0.02) | 0.01 (0.01) | *2.55 (0.40)* | *2.91 (0.48)* | *3.32 (0.62)* |
| 750 | | 0.77 (0.10) | 0.14 (0.03) | 0.06 (0.02) | *1.97 (0.24)* | *2.13 (0.25)* | *2.38 (0.31)* |
| 1000 | | *1.42 (0.11)* | 0.26 (0.03) | 0.07 (0.01) | *2.01 (0.16)* | *1.57 (0.13)* | *1.61 (0.14)* |

| 1250 | | *2.03 (0.18)* | 0.53 (0.08) | 0.07 (0.01) | *2.38 (0.21)* | *1.47 (0.21)* | **1.22 (0.21)** |
|---|---|---|---|---|---|---|---|
| 250 | | 0.29 (0.12) | 0.00 (0.00) | 0.00 (0.00) | *4.05 (0.95)* | *4.11 (1.04)* | *4.22 (1.26)* |
| 500 | | 0.44 (0.08) | 0.06 (0.02) | 0.02 (0.02) | *3.18 (0.52)* | *3.51 (0.59)* | *4.07 (0.77)* |
| 750 | 15 | 0.75 (0.08) | 0.18 (0.03) | 0.04 (0.01) | *1.85 (0.17)* | *1.89 (0.20)* | *2.22 (0.24)* |
| 1000 | | *1.47 (0.14)* | 0.29 (0.04) | 0.05 (0.02) | *2.06 (0.20)* | *1.46 (0.19)* | **1.38 (0.21)** |
| 1250 | | *2.32 (0.16)* | 0.72 (0.09) | 0.08 (0.02) | *2.67 (0.19)* | *1.87 (0.21)* | **1.34 (0.23)** |

*Bold* indicates that $\rho$ values are not statistically significantly different than one (i.e., no safety changes between the 2+1 and the TLTW designs) at the 5% significance level. *Italic* (Red) indicates that $\rho$ values are statistically significant >1.0 (i.e., reduced safety of the 2+1 design compared to the TLTW design) at the 5% significance level. *Neither Bold nor Italic* indicates that $\rho$ values are statistically significant <1.0 (i.e., improved safety of the 2+1 design compared to the TLTW design) at the 5% significance level.

**Table 8.** Conflict ratio, with standard error between conventional TLTW before and after 2+1 design for network speed of 70 km/h for USL Strategy.

| V | %HV | TLTW to 2+1 (No Pass) | | | TLTW to 2+1 (Allow Pass) | | |
|---|---|---|---|---|---|---|---|
| | | TTC ≤ 2.50 | TTC ≤ 1.50 | TTC ≤ 0.50 | TTC ≤ 2.50 | TTC ≤ 1.50 | TTC ≤ 0.50 |
| 250 | | 0.12 (0.06) | 0.00 (0.00) | 0.00 (0.00) | *9.38 (2.25)* | *9.70 (2.43)* | *10.40 (2.88)* |
| 500 | | 0.22 (0.03) | 0.06 (0.01) | 0.03 (0.01) | *5.98 (0.74)* | *6.84 (0.86)* | *7.69 (1.07)* |
| 750 | 2.5 | 0.45 (0.05) | 0.11 (0.02) | 0.02 (0.01) | *3.90 (0.37)* | *4.85 (0.49)* | *5.32 (0.59)* |
| 1000 | | 0.95 (0.09) | 0.22 (0.03) | 0.08 (0.02) | *3.25 (0.28)* | *3.72 (0.33)* | *4.13 (0.45)* |
| 1250 | | 2.00 (0.22) | 0.43 (0.05) | 0.11 (0.02) | *3.47 (0.38)* | *2.72 (0.27)* | *2.99 (0.38)* |
| 250 | | 0.10 (0.04) | 0.02 (0.02) | 0.00 (0.00) | *9.69 (2.34)* | *10.35 (2.50)* | *10.39 (2.30)* |
| 500 | | 0.21 (0.03) | 0.05 (0.01) | 0.03 (0.01) | *6.20 (0.50)* | *7.25 (0.56)* | *8.11 (0.77)* |
| 750 | 5 | 0.41 (0.05) | 0.11 (0.02) | 0.02 (0.01) | *3.63 (0.37)* | *4.44 (0.43)* | *5.22 (0.57)* |
| 1000 | | **1.03 (0.07)** | 0.24 (0.03) | 0.09 (0.01) | *3.07 (0.21)* | *3.52 (0.24)* | *3.85 (0.30)* |
| 1250 | | *1.69 (0.12)* | 0.33 (0.03) | 0.08 (0.01) | *2.83 (0.19)* | *2.29 (0.15)* | *2.30 (0.16)* |
| 250 | | 0.09 (0.06) | 0.00 (0.00) | 0.00 (0.00) | *10.28 (1.62)* | *10.86 (1.65)* | *12.05 (2.03)* |
| 500 | | 0.26 (0.04) | 0.05 (0.01) | 0.01 (0.01) | *5.93 (0.57)* | *6.83 (0.66)* | *7.70 (0.89)* |
| 750 | 10 | 0.47 (0.04) | 0.09 (0.01) | 0.02 (0.01) | *3.80 (0.29)* | *4.73 (0.40)* | *5.52 (0.48)* |
| 1000 | | **1.07 (0.13)** | 0.25 (0.03) | 0.07 (0.02) | *3.05 (0.36)* | *3.58 (0.41)* | *4.02 (0.52)* |
| 1250 | | *1.83 (0.11)* | 0.33 (0.04) | 0.08 (0.01) | *2.86 (0.19)* | *2.17 (0.17)* | *2.25 (0.19)* |
| 250 | | 0.03 (0.02) | 0.02 (0.02) | 0.00 (0.00) | *8.42 (1.44)* | *9.05 (1.56)* | *11.84 (2.65)* |
| 500 | | 0.28 (0.04) | 0.05 (0.01) | 0.02 (0.01) | *5.97 (0.40)* | *6.66 (0.43)* | *7.53 (0.66)* |
| 750 | 15 | 0.68 (0.08) | 0.11 (0.02) | 0.03 (0.01) | *4.53 (0.43)* | *5.20 (0.52)* | *5.84 (0.57)* |
| 1000 | | *1.20 (0.10)* | 0.27 (0.03) | 0.07 (0.01) | *3.31 (0.29)* | *3.72 (0.31)* | *4.02 (0.42)* |
| 1250 | | *2.01 (0.24)* | 0.35 (0.05) | 0.07 (0.02) | *2.94 (0.36)* | *2.16 (0.29)* | *2.15 (0.32)* |

*Bold* indicates that $\rho$ values are not statistically significantly different than one (i.e., no safety changes between the 2+1 and the TLTW designs) at the 5% significance level. *Italic* (Red) indicates that $\rho$ values are statistically significant >1.0 (i.e., reduced safety of the 2+1 design compared to the TLTW design) at the 5% significance level. *Neither Bold nor Italic* indicates that $\rho$ values are statistically significant <1.0 (i.e., improved safety of the 2+1 design compared to the TLTW design) at the 5% significance level.

**Table 9.** Conflict ratio, with standard error between conventional TLTW before and after 2+1 design for network speed of 80 km/h for USL Strategy.

| V | % HV | TLTW to 2+1 (No Pass) | | | TLTW to 2+1 (Allow Pass) | | |
|---|---|---|---|---|---|---|---|
| | | TTC ≤ 2.50 | TTC ≤ 1.50 | TTC ≤ 0.50 | TTC ≤ 2.50 | TTC ≤ 1.50 | TTC ≤ 0.50 |
| 250 | | 0.08 (0.04) | 0.03 (0.02) | 0.00 (0.00) | *6.97 (1.37)* | *7.16 (1.42)* | *6.70 (1.25)* |
| 500 | | 0.13 (0.02) | 0.05 (0.01) | 0.03 (0.01) | *4.66 (0.51)* | *4.93 (0.53)* | *5.87 (0.75)* |
| 750 | 2.5 | 0.29 (0.03) | 0.12 (0.02) | 0.04 (0.01) | *2.92 (0.25)* | *3.31 (0.26)* | *3.76 (0.39)* |
| 1000 | | 0.76 (0.09) | 0.30 (0.04) | 0.11 (0.02) | *2.42 (0.28)* | *2.40 (0.27)* | *2.60 (0.33)* |
| 1250 | | *1.47 (0.15)* | 0.54 (0.06) | 0.22 (0.04) | *2.42 (0.25)* | *1.87 (0.21)* | *1.86 (0.23)* |
| 250 | | 0.11 (0.05) | 0.05 (0.03) | 0.00 (0.00) | *7.16 (1.13)* | *7.25 (1.13)* | *7.82 (1.18)* |
| 500 | | 0.16 (0.02) | 0.05 (0.02) | 0.03 (0.01) | *5.46 (0.56)* | *6.19 (0.60)* | *7.18 (0.89)* |
| 750 | 5 | 0.40 (0.05) | 0.16 (0.02) | 0.06 (0.01) | *3.33 (0.37)* | *3.58 (0.36)* | *4.09 (0.46)* |
| 1000 | | 0.75 (0.05) | 0.27 (0.03) | 0.11 (0.01) | *2.29 (0.14)* | *2.31 (0.16)* | *2.53 (0.20)* |
| 1250 | | *1.58 (0.13)* | 0.51 (0.05) | 0.18 (0.02) | *2.54 (0.20)* | *1.91 (0.14)* | *1.83 (0.15)* |
| 250 | | 0.06 (0.03) | 0.03 (0.02) | 0.00 (0.00) | *7.31 (0.79)* | *7.25 (0.75)* | *8.57 (1.42)* |
| 500 | | 0.23 (0.03) | 0.06 (0.01) | 0.02 (0.01) | *5.18 (0.51)* | *5.60 (0.50)* | *6.81 (0.68)* |
| 750 | 10 | 0.40 (0.05) | 0.16 (0.02) | 0.05 (0.01) | *3.15 (0.38)* | *3.42 (0.36)* | *4.01 (0.47)* |
| 1000 | | **0.89 (0.08)** | 0.30 (0.03) | 0.10 (0.01) | *2.44 (0.22)* | *2.28 (0.23)* | *2.45 (0.27)* |
| 1250 | | *2.04 (0.22)* | 0.59 (0.07) | 0.22 (0.04) | *2.93 (0.32)* | *1.91 (0.20)* | *1.84 (0.22)* |
| 250 | | 0.07 (0.04) | 0.04 (0.03) | 0.02 (0.02) | *8.46 (1.64)* | *8.45 (1.66)* | *8.62 (1.98)* |
| 500 | | 0.26 (0.03) | 0.06 (0.01) | 0.01 (0.01) | *5.37 (0.54)* | *5.53 (0.53)* | *6.53 (0.76)* |
| 750 | 15 | 0.36 (0.03) | 0.13 (0.01) | 0.04 (0.01) | *2.63 (0.20)* | *2.85 (0.18)* | *3.20 (0.22)* |
| 1000 | | **0.86 (0.08)** | 0.28 (0.03) | 0.12 (0.02) | *2.17 (0.21)* | *2.04 (0.20)* | *2.18 (0.24)* |
| 1250 | | *1.99 (0.30)* | 0.63 (0.06) | 0.27 (0.03) | *3.11 (0.47)* | *2.18 (0.17)* | *2.30 (0.21)* |

*Bold* indicates that $\rho$ values are not statistically significantly different than one (i.e., no safety changes between the 2+1 and the TLTW designs) at the 5% significance level. *Italic* (Red) indicates that $\rho$ values are statistically significant >1.0 (i.e., reduced safety of the 2+1 design compared to the TLTW design) at the 5% significance level. *Neither Bold nor Italic* indicates that $\rho$ values are statistically significant <1.0 (i.e., improved safety of the 2+1 design compared to the TLTW design) at the 5% significance level.

**Table 10.** Conflict ratio, with standard error between conventional TLTW before and after 2+1 design for network speed of 90 km/h for USL Strategy.

| V | %HV | TLTW to 2+1 (No Pass) | | | TLTW to 2+1 (Allow Pass) | | |
|---|---|---|---|---|---|---|---|
| | | TTC ≤ 2.50 | TTC ≤ 1.50 | TTC ≤ 0.50 | TTC ≤ 2.50 | TTC ≤ 1.50 | TTC ≤ 0.50 |
| 250 | | 0.00 (0.00) | 0.00 (0.00) | 0.00 (0.00) | *5.63 (0.88)* | *5.58 (0.88)* | *5.87 (1.08)* |
| 500 | | 0.14 (0.04) | 0.08 (0.02) | 0.04 (0.01) | *4.14 (0.40)* | *4.21 (0.39)* | *5.09 (0.51)* |
| 750 | 2.5 | 0.34 (0.05) | 0.19 (0.03) | 0.09 (0.02) | *2.60 (0.39)* | *2.75 (0.40)* | *3.20 (0.57)* |
| 1000 | | 0.68 (0.07) | 0.35 (0.03) | 0.14 (0.02) | *1.96 (0.17)* | *1.82 (0.16)* | *1.90 (0.19)* |
| 1250 | | 1.72 (0.15) | 0.78 (0.08) | 0.30 (0.03) | *2.53 (0.24)* | *1.75 (0.18)* | *1.51 (0.16)* |
| 250 | | 0.00 (0.00) | 0.00 (0.00) | 0.00 (0.00) | *4.57 (0.75)* | *4.55 (0.75)* | *5.79 (1.11)* |
| 500 | | 0.19 (0.03) | 0.09 (0.02) | 0.04 (0.01) | *3.93 (0.36)* | *4.26 (0.41)* | *4.85 (0.57)* |
| 750 | 5 | 0.36 (0.05) | 0.19 (0.03) | 0.08 (0.02) | *2.53 (0.28)* | *2.57 (0.30)* | *3.11 (0.44)* |
| 1000 | | 0.56 (0.04) | 0.27 (0.02) | 0.12 (0.02) | *1.50 (0.12)* | *1.41 (0.12)* | *1.47 (0.14)* |
| 1250 | | 1.95 (0.20) | **0.94 (0.11)** | 0.36 (0.05) | *2.72 (0.28)* | *1.96 (0.23)* | *1.64 (0.21)* |
| 250 | | 0.01 (0.01) | 0.00 (0.00) | 0.00 (0.00) | *4.40 (0.76)* | *4.42 (0.75)* | *4.51 (0.87)* |
| 500 | | 0.14 (0.04) | 0.08 (0.02) | 0.03 (0.01) | *4.28 (0.42)* | *4.37 (0.44)* | *4.94 (0.48)* |
| 750 | 10 | 0.32 (0.03) | 0.18 (0.02) | 0.08 (0.01) | *2.25 (0.20)* | *2.31 (0.21)* | *2.73 (0.25)* |
| 1000 | | 0.81 (0.06) | 0.38 (0.03) | 0.16 (0.02) | *1.86 (0.14)* | *1.66 (0.14)* | *1.78 (0.17)* |
| 1250 | | 2.02 (0.11) | 0.87 (0.06) | 0.32 (0.03) | *2.72 (0.14)* | *1.81 (0.09)* | *1.46 (0.10)* |
| 250 | 15 | 0.01 (0.02) | 0.00 (0.00) | 0.00 (0.00) | *4.92 (0.91)* | *4.88 (0.90)* | *5.52 (1.16)* |

| 500 | 0.17 (0.03) | 0.07 (0.02) | 0.03 (0.01) | *3.76 (0.47)* | *3.91 (0.45)* | *4.29 (0.57)* |
| 750 | 0.39 (0.04) | 0.19 (0.02) | 0.10 (0.02) | *2.14 (0.20)* | *2.23 (0.20)* | *2.55 (0.25)* |
| 1000 | 0.98 (0.07) | 0.43 (0.04) | 0.15 (0.02) | *2.05 (0.16)* | *1.74 (0.17)* | *1.73 (0.18)* |
| 1250 | 2.07 (0.16) | **0.85 (0.08)** | 0.31 (0.04) | *2.67 (0.21)* | *1.68 (0.16)* | **1.27 (0.14)** |

*Bold* indicates that $\rho$ values are not statistically significantly different than one (i.e., no safety changes between the 2+1 and the TLTW designs) at the 5% significance level. *Italic* (Red) indicates that $\rho$ values are statistically significant >1.0 (i.e., reduced safety of the 2+1 design compared to the TLTW design) at the 5% significance level. *Neither Bold nor Italic* indicates that $\rho$ values are statistically significant <1.0 (i.e., improved safety of the 2+1 design compared to the TLTW design) at the 5% significance level.

### 4.2. Differential Speed-Limit (DSL) Results

#### 4.2.1. Simulated Mobility Results

Figures 5 and 6 and Tables 11 and 12 show the average delay and the ATS results for the differential speed-limit scenarios, for speed limits of 80 and 90 km/h for PCs and 60 km/h for HVs for different traffic volumes and different percentages of HVs, for both of the conventional TLTW, and the 2+1 that prohibits overtaking in the opposite direction designs. It is worth noting that the 2+1 design that allows overtaking in the opposite direction was not further investigated, as the USL scenarios showed that this design compromised safety compared to the TLTW design.

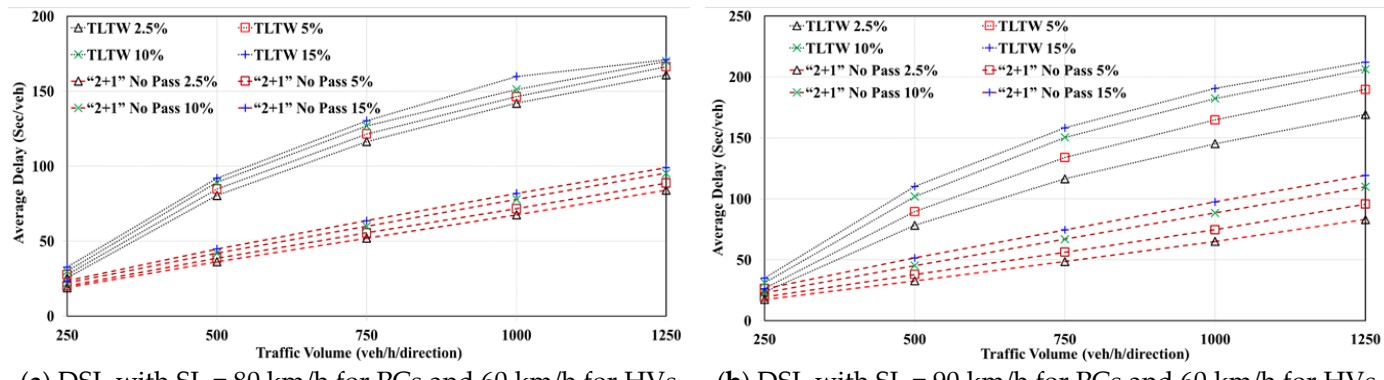

(**a**) DSL with SL = 80 km/h for PCs and 60 km/h for HVs     (**b**) DSL with SL = 90 km/h for PCs and 60 km/h for HVs

**Figure 5.** Average network delay (second/vehicle) for differential speed-limit scenarios: (**a**) speed limit = 80 km/h for PCs and 60 km/h for HVs, (**b**) speed limit = 90 km/h for PCs and 60 km/h for HVs.

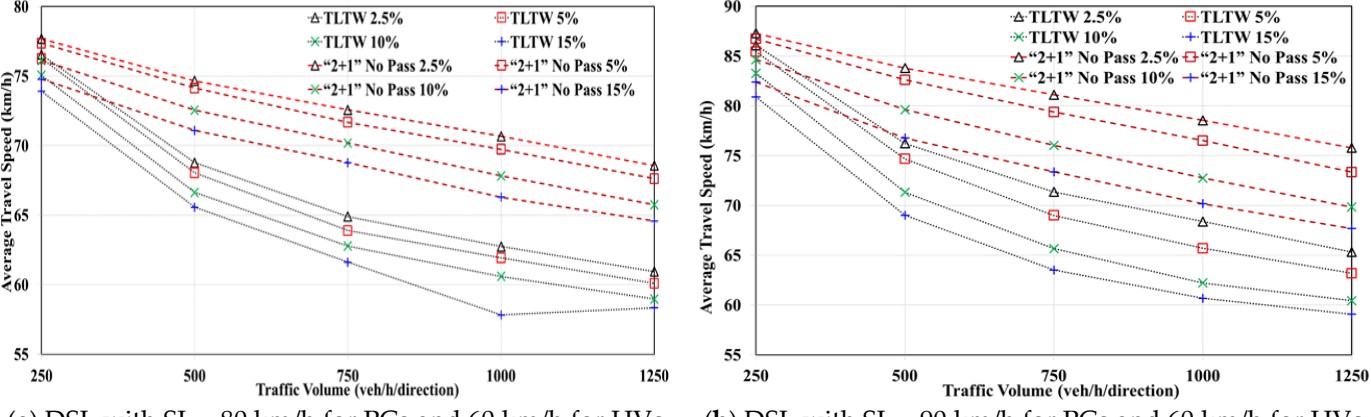

(**a**) DSL with SL = 80 km/h for PCs and 60 km/h for HVs     (**b**) DSL with SL = 90 km/h for PCs and 60 km/h for HVs

**Figure 6.** Average travel speed (km/h) for differential speed-limit (DSL) scenarios: (**a**) speed limit = 80 km/h for PCs and 60 km/h for HVs, (**b**) speed limit = 90 km/h for PCs and 60 km/h for HVs.

**Table 11.** The average travel speed (km/h) and the average delay (sec/veh) results for speed limit of 80 km/h for PCs and 60 km/h for HVs.

| V1 | %HV | TLTW | | 2+1 (No Pass) | |
|---|---|---|---|---|---|
| | | Speed (Km/h) | Delay (S/veh) | Speed (Km/h) | Delay (S/veh) |
| 250 | | 76.54 ± 0.31 | 25.54 ± 1.97 | 77.68 ± 0.33 (1.5) | 18.96 ± 1.41 (−26) |
| 500 | | 68.78 ± 0.26 | 80.48 ± 3.03 | 74.66 ± 0.23 (8.6) | 36.32 ± 1.33 (−55) |
| 750 | 2.5 | 64.89 ± 0.23 | 116.45 ± 3.24 | 72.57 ± 0.22 (11.8) | 52.03 ± 1.63 (−55) |
| 1000 | | 62.74 ± 0.21 | 142.07 ± 3.09 | 70.67 ± 0.23 (12.6) | 67.63 ± 1.93 (−52) |
| 1250 | | 60.94 ± 0.14 | 160.91 ± 1.96 | 68.57 ± 0.17 (12.5) | 84.07 ± 1.40 (−48) |
| 250 | | 76.25 ± 0.36 | 27.67 ± 1.94 | 77.36 ± 0.34 (1.5) | 20.14 ± 1.37 (−27) |
| 500 | | 68.04 ± 0.29 | 84.76 ± 3.65 | 74.14 ± 0.23 (9.0) | 38.67 ± 1.23 (−54) |
| 750 | 5 | 63.90 ± 0.21 | 121.50 ± 3.35 | 71.67 ± 0.20 (12.2) | 55.54 ± 1.53 (−54) |
| 1000 | | 61.93 ± 0.16 | 146.40 ± 2.77 | 69.73 ± 0.20 (12.6) | 71.66 ± 1.86 (−51) |
| 1250 | | 60.10 ± 0.15 | 166.36 ± 3.55 | 67.62 ± 0.18 (12.5) | 88.78 ± 1.54 (−47) |
| 250 | | 75.05 ± 0.34 | 30.00 ± 1.86 | 76.12 ± 0.30 (1.4) | 22.10 ± 1.02 (−26) |
| 500 | | 66.63 ± 0.23 | 89.16 ± 2.90 | 72.54 ± 0.19 (8.9) | 41.96 ± 0.90 (−53) |
| 750 | 10 | 62.77 ± 0.15 | 126.89 ± 2.74 | 70.18 ± 0.19 (11.8) | 59.88 ± 1.33 (−53) |
| 1000 | | 60.59 ± 0.10 | 151.23 ± 1.94 | 67.82 ± 0.17 (11.9) | 77.86 ± 1.46 (−49) |
| 1250 | | 58.97 ± 0.12 | 170.00 ± 2.35 | 65.74 ± 0.16 (11.5) | 95.37 ± 1.56 (−44) |
| 250 | | 73.90 ± 0.41 | 32.75 ± 2.78 | 74.78 ± 0.36 (1.2) | 23.49 ± 1.25 (−28) |
| 500 | | 65.57 ± 0.27 | 91.99 ± 3.91 | 71.08 ± 0.19 (8.4) | 44.79 ± 1.04 (−51) |
| 750 | 15 | 61.62 ± 0.13 | 130.40 ± 2.37 | 68.77 ± 0.16 (11.6) | 63.67 ± 1.51 (−51) |
| 1000 | | 57.82 ± 0.37 | 159.86 ± 12.86 | 66.28 ± 0.18 (14.6) | 81.90 ± 1.59 (−49) |
| 1250 | | 58.34 ± 0.12 | 171.04 ± 2.47 | 64.58 ± 0.17 (10.7) | 99.11 ± 1.86 (−42) |

**Table 12.** The average travel speed (km/h) and the average delay (sec/veh) results for speed limit of 90 km/h for PCs and 60 km/h for HVs.

| V | %HV | TLTW | | 2+1 (No Pass) | |
|---|---|---|---|---|---|
| | | Speed | Delay | Speed | Delay |
| 250 | | 86.08 ± 0.42 | 23.75 ± 1.61 | 87.29 ± 0.38 (1.4) | 17.51 ± 1.30 (−26) |
| 500 | | 76.23 ± 0.38 | 78.39 ± 4.37 | 83.79 ± 0.25 (9.9) | 32.82 ± 0.95 (−58) |
| 750 | 2.5 | 71.36 ± 0.37 | 116.58 ± 4.33 | 81.12 ± 0.31 (13.7) | 48.64 ± 1.92 (−58) |
| 1000 | | 68.38 ± 0.36 | 145.26 ± 4.83 | 78.55 ± 0.34 (14.9) | 65.09 ± 2.85 (−55) |
| 1250 | | 65.33 ± 0.29 | 169.24 ± 4.58 | 75.78 ± 0.26 (16.0) | 83.11 ± 2.42 (−51) |
| 250 | | 85.44 ± 0.50 | 26.62 ± 2.24 | 86.73 ± 0.44 (1.5) | 19.54 ± 1.48 (−27) |
| 500 | | 74.67 ± 0.41 | 89.64 ± 4.59 | 82.61 ± 0.29 (10.6) | 37.97 ± 1.47 (−58) |
| 750 | 5 | 69.00 ± 0.29 | 134.02 ± 3.98 | 79.38 ± 0.25 (15.1) | 56.25 ± 1.42 (−58) |
| 1000 | | 65.70 ± 0.38 | 164.77 ± 7.15 | 76.52 ± 0.27 (16.5) | 74.76 ± 2.14 (−55) |
| 1250 | | 63.19 ± 0.32 | 189.86 ± 5.70 | 73.34 ± 0.29 (16.1) | 95.69 ± 2.91 (−50) |
| 250 | | 83.26 ± 0.49 | 31.25 ± 2.11 | 84.64 ± 0.42 (1.7) | 22.97 ± 1.53 (−26) |
| 500 | | 71.33 ± 0.25 | 101.97 ± 2.92 | 79.62 ± 0.26 (11.6) | 45.32 ± 1.58 (−56) |
| 750 | 10 | 65.66 ± 0.26 | 150.45 ± 4.27 | 76.04 ± 0.25 (15.8) | 66.95 ± 1.68 (−56) |
| 1000 | | 62.22 ± 0.22 | 182.35 ± 4.26 | 72.75 ± 0.23 (16.9) | 88.55 ± 1.84 (−51) |
| 1250 | | 60.44 ± 0.24 | 206.45 ± 4.77 | 69.84 ± 0.26 (15.6) | 109.97 ± 2.54 (−47) |
| 250 | | 80.89 ± 0.56 | 35.09 ± 2.72 | 82.38 ± 0.50 (1.8) | 26.34 ± 1.38 (−25) |
| 500 | | 69.01 ± 0.38 | 110.32 ± 5.19 | 76.78 ± 0.29 (11.2) | 51.68 ± 1.90 (−53) |
| 750 | 15 | 63.51 ± 0.29 | 158.38 ± 4.45 | 73.38 ± 0.26 (15.5) | 74.58 ± 2.00 (−53) |
| 1000 | | 60.68 ± 0.19 | 190.63 ± 3.79 | 70.16 ± 0.26 (15.6) | 97.60 ± 2.40 (−49) |
| 1250 | | 59.07 ± 0.18 | 212.25 ± 3.16 | 67.68 ± 0.29 (14.6) | 119.29 ± 3.23 (−44) |

In contrast to the USL strategy, the DSL strategy shows that by increasing the portion of heavy vehicles in the traffic stream, there is a significant decrease in delay and a significant increase in the ATS for all speed limits and all traffic volumes. In addition, for traffic volumes greater than 500 vehicle/h/direction, the results of the DSL scenarios show that the 2+1 design would significantly decrease the average network delay compared to the TLTW design. The reduction in the average delay is between 26% and 48% and between 26% and 51% for speed limits of 80 km/h and 90 km/h, respectively. Furthermore, the results of the DSL scenarios show that the 2+1 design would significantly increase the ATS compared to the TLTW design. The increase in the ATS is from 1.5% to 12.5% and from 1.4% to 16.9% for speed limits of 80 km/h and 90 km/h, respectively, with approximately the same rate of increase for all percentages of HVs. However, for a traffic volume of 250 vehicle/h/direction (i.e., free-flow condition), there is no statistically significant difference in the ATS for both the TLTW and the 2+1 designs.

It is worth noting that, for all the USL scenarios (Figure 4) and all DSL scenarios (Figure 6), there is a noticeable reduction in the ATS with the increase in traffic volume. In addition, the results show that, by converting the 6.0-km segment from TLTW to 2+1, the ATS would increase for all traffic volumes. This proves the mobility efficiency of the 2+1 design compared to the TLTW design.

### 4.2.2. Simulated Safety Results

Table 13 shows the conflict ratio ($\rho$) for the total simulated conflicts for different TTC thresholds for the USL strategy, with speed limits of 80 and 90 km/h for PCs and 60 km/h for HVs. The results in Table 13 show that the $\rho$ value for almost all scenarios (with both speed limits with different traffic volumes) with different TTC thresholds is less than 1.0, which indicates a reduction in the simulated conflicts and, hence, a reduction in crashes. There are only a few values at TTC ≤ 2.50 s that show a statistically significant value of $\rho$ > 1.0 at a traffic volume of 1250 vehicle/h, which indicates an increase in the simulated conflicts. These instances should not be viewed as safety drops, as mentioned earlier.

**Table 13.** Conflict ratio between conventional TLTW before and after 2+1 (no pass) design with the DSL Strategy.

| V | %HV | SL = 80 km/h for PCs and 60 km/h for HVs | | | SL = 90 km/h for PCs and 60 km/h for HVs | | |
| --- | --- | --- | --- | --- | --- | --- | --- |
| | | TTC ≤ 2.50 | TTC ≤ 1.50 | TTC ≤ 0.50 | TTC ≤ 2.50 | TTC ≤ 1.50 | TTC ≤ 0.50 |
| 250 | | 0.09 (0.04) | 0.00 (0.00) | 0.00 (0.00) | 0.05 (0.02) | 0.01 (0.01) | 0.01 (0.01) |
| 500 | | 0.17 (0.03) | 0.02 (0.01) | 0.02 (0.01) | 0.12 (0.02) | 0.03 (0.01) | 0.01 (0.00) |
| 750 | 2.5 | 0.38 (0.03) | 0.11 (0.01) | 0.04 (0.01) | 0.27 (0.03) | 0.09 (0.01) | 0.04 (0.01) |
| 1000 | | 0.73 (0.08) | 0.19 (0.02) | 0.07 (0.01) | 0.48 (0.06) | 0.16 (0.02) | 0.07 (0.01) |
| 1250 | | *1.40 (0.16)* | 0.32 (0.04) | 0.10 (0.02) | **0.94 (0.07)** | 0.30 (0.02) | 0.10 (0.01) |
| 250 | | 0.06 (0.02) | 0.01 (0.01) | 0.00 (0.00) | 0.04 (0.01) | 0.01 (0.01) | 0.00 (0.00) |
| 500 | | 0.26 (0.03) | 0.04 (0.01) | 0.02 (0.01) | 0.15 (0.02) | 0.03 (0.01) | 0.01 (0.00) |
| 750 | 5 | 0.33 (0.03) | 0.06 (0.01) | 0.02 (0.00) | 0.26 (0.03) | 0.05 (0.01) | 0.02 (0.00) |
| 1000 | | 0.75 (0.07) | 0.14 (0.02) | 0.04 (0.01) | 0.54 (0.03) | 0.13 (0.01) | 0.04 (0.00) |
| 1250 | | *1.45 (0.11)* | 0.29 (0.03) | 0.09 (0.01) | **0.99 (0.06)** | 0.22 (0.01) | 0.07 (0.01) |
| 250 | | 0.13 (0.04) | 0.02 (0.01) | 0.00 (0.00) | 0.07 (0.01) | 0.00 (0.00) | 0.00 (0.00) |
| 500 | | 0.24 (0.02) | 0.03 (0.00) | 0.01 (0.00) | 0.15 (0.01) | 0.02 (0.01) | 0.01 (0.00) |
| 750 | 10 | 0.38 (0.03) | 0.05 (0.01) | 0.01 (0.00) | 0.29 (0.01) | 0.05 (0.00) | 0.02 (0.00) |
| 1000 | | 0.82 (0.06) | 0.11 (0.02) | 0.03 (0.01) | 0.58 (0.04) | 0.09 (0.01) | 0.03 (0.00) |
| 1250 | | *1.95 (0.19)* | 0.29 (0.03) | 0.06 (0.01) | *1.13 (0.06)* | 0.18 (0.01) | 0.05 (0.01) |
| 250 | | 0.20 (0.04) | 0.01 (0.01) | 0.00 (0.00) | 0.09 (0.01) | 0.00 (0.00) | 0.00 (0.00) |
| 500 | | 0.26 (0.03) | 0.02 (0.00) | 0.01 (0.00) | 0.16 (0.01) | 0.02 (0.00) | 0.01 (0.00) |
| 750 | 15 | 0.39 (0.03) | 0.04 (0.01) | 0.01 (0.00) | 0.33 (0.02) | 0.04 (0.00) | 0.02 (0.00) |
| 1000 | | 0.83 (0.04) | 0.10 (0.01) | 0.03 (0.00) | 0.63 (0.03) | 0.08 (0.01) | 0.02 (0.00) |
| 1250 | | *2.67 (0.21)* | 0.31 (0.03) | 0.06 (0.01) | *1.48 (0.08)* | 0.19 (0.01) | 0.04 (0.00) |

Bold indicates that ϱ values are not statistically significantly different than 1.0 (i.e., no safety changes between the 2+1 and the TLTW designs) at the 5% significance level. Italic (red) indicates that ϱ values are statistically significant >1.0 (i.e., reduced safety of the 2+1 design compared to the TLTW design) at the 5% significance level. Neither bold nor italic indicates that ϱ values are statistically

significant <1.0 (i.e., improved safety of the 2+1 design compared to the TLTW design) at the 5% significance level.

## 5. Conclusions

This study presents an evaluation of the 2+1 road design versus the conventional TLTW design, for Middle Eastern conditions. It initiates with the utilization of the SUMO traffic microscopic simulation model to model the movements of passenger cars (PCs) and heavy vehicles (HVs) within the highway network over a designated timeframe. The calibrated SUMO parameters were based on a segment of a TLTW from Egypt. In our analysis, we used a 6.0-kilometer-long road with a lane width of 3.65 m for both designs (i.e., TLTW and 2+1) and an auxiliary passing lane of 1.0 kilometer and a lane width of 3.65 m. In addition, we tested five different speed limits, namely 50, 60, 70, 80, and 90 km/h, and five different traffic volumes (i.e., 250 to 1250 vehicle/hour/direction). Furthermore, the heavy-vehicle effects were explored at four different levels, namely 2.50%, 5.0%, 10.0%, and 15%, of the total traffic volume. Moreover, two speed-limit strategies (i.e., USL and DSL) were investigated, as well. The average network delay and the average travel speed (ATS) were used as the indicators for the mobility performance, while the number of simulated traffic conflicts, based on the time to collision (TTC), was used as the surrogate safety indicator to reflect the crash potential for prohibited overtaking and allowed overtaking in the opposite direction in the 2+1 design. The main conclusions based on the mobility performance assessment are as follows.

- For different USLs, the speed improvement was between 1.4% (at low traffic volumes) and 12.5% (at traffic volumes near capacity) and between 3.3% and 13.80% for the 2+1 design that prohibits overtaking and that allows it, respectively. And, the reduction in the average network delay varies from 26% to 50% and from 47% to 74% for the 2+1 design that prohibits overtaking and that allows it, respectively. While the presence of heavy vehicles has a negligible effect on traffic mobility;
- For the DSL strategy, as the HVs percentage increases in the traffic stream, there is a significant change in the ATS and the network delay. The reduction in the average delay was between 26% and 48% and between 26% and 51% for speed limits of 80 km/h and 90 km/h, respectively. In addition, the increase in the ATS was from 1.5% to 12.5% and from 1.4% to 16.9% for speed limits of 80 km/h and 90 km/h, respectively, with approximately the same rate of increase for all % HVs.

Furthermore, the main conclusions based on the safety assessment are as follows.

- For the 2+1 design that prohibits overtaking in the opposite direction, the simulated conflict ratio ($\rho$ value) for almost all scenarios, with both the USL and DSL speed-limit strategies, different traffic volumes, and different TTC thresholds, was less than "1.0", which suggests a reduction in the simulated conflicts and, hence, a reduction in the expected crashes compared to the TLTW design. This suggests a safety improvement that would be expected when converting the TLTW roads to 2+1 roads and prohibiting vehicles from overtaking in opposite directions by using an appropriate median barrier;
- For the 2+1 design that allows overtaking in the opposite direction, the $\rho$ value for different traffic volumes and speed limits was greater than "1.0", which indicates an increase in the simulated conflicts and, hence, an increase in the expected crashes compared to the TLTW design. Therefore, allowing overtaking in the opposite direction is not recommended when converting a TLTW road to a 2+1 road.

The results provide valuable information to policymakers, urban planners, and transport authorities to guide evidence-based decisions on the integration of 2+1 design as a viable solution for sustainable and efficient transportation. In future work, other configurations for the length of the passing lane may be studied. Furthermore, a comparison between the 2+1 design and the 2+2 design would be beneficial for assessing the conver-

sion of current TLTW (i.e., 1+1 design) roads to 2+1 roads as an intermediate solution before upgrading to the 2+2 (i.e., 4-lane highways) design. Additionally, a cost–benefit analysis can be estimated for all proven effective scenarios. Finally, proper maintenance and enforcement are essential to ensure the effectiveness of the 2+1 design, including the regular inspection of barriers, signage, and road markings, as well as the enforcement of speed limits and overtaking rules.

**Author Contributions:** Conceptualization, U.E.S. and M.E.; methodology, U.E.S., F. A., A. A. and M.E.; software, U.E.S.; validation, U.E.S., M.E., A.A. and F.A.; formal analysis, U.E.S., F. F., A. A.; data curation, U.E.S.; writing—original draft preparation, U.E.S., F. F., A.A. and M.E.; writing—review and editing, U.E.S., F.A., A.A., and M.E.; project administration, F.A. and A.A. All authors have read and agreed to the published version of the manuscript.

**Funding:** This research received no external funding.

**Institutional Review Board Statement:** Not applicable.

**Informed Consent Statement:** Not applicable.

**Data Availability Statement:** The data presented in this study are available on request from the corresponding author.

**Acknowledgments:** The authors would like to express their gratitude to Ahmed Shoaeb, Graduate Student at the Public Works Engineering Department, Faculty of Engineering, Mansoura University, for providing the data used in calibrating the SUMO software [33].

**Conflicts of Interest:** The authors declare no conflicts of interest.

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
