# Peer review of "Safety and Mobility Performance Comparison of Two-Plus-One and Two-Lane Two-Way Roads: A Simulation Study"

_applsci, doi:10.3390/app14114352_

Round 1

Reviewer 1 Report

Comments and Suggestions for Authors

General comments and suggestions for the Author:

The present investigates the “Safety and Mobility Performance Comparison of Two-Plus-One and Two-Lane Two-Way Roads: A Simulation Study”. The topic is interesting and important. However, there are several key areas that need more work prior to publication.

      i.         The overall manuscript requires substantial editing and revising to correct the numerous language and grammar errors. The authors should ask the help of native English speaking, proofreader, because there are some typo and linguistic mistakes that should be fixed.

     ii.         Some assumptions are stated in various sections. Justifications should be provided on these assumptions. Evaluation on how they will affect the results should be made.

    iii.         Please add a sentence or two to clearly recap how your study differs from what has already been done in literature to ascertain the contributions more strongly.

    iv.         Analysis, elaboration, and presentation of this work need to be improved before it can be considered for publication. For improved clarity, consider utilizing distinct symbols in graphs for easier identification. Additionally, plotting percentage levels on separate graphs would facilitate differentiation between data for Two-Lane Two-Way Roads (TLTW) and Two-Plus-One roads.

Please read through my comments below and incorporate them into your manuscript to improve its quality and readability. The comments are listed below:

Introduction

The introduction needs to be revised. Also, it does not properly refer to previously published studies. The authors need to carefully review the published literature, identify the gaps in the literature, and propose their approach to fill the gap. It is important to also add some recent work (2023-2024) to the literature review. At least 5 new references should be added to article.

Consider the following:

Lines 37: Define “TWTL” as this is the first time it appears in the manuscript.

Lines 44 to 45: the authors state that “Transportation policies have historically focused on infrastructure development, aiming to improve road networks and enhance public transportation systems. where does this come from? Where is the evidence? reference missing?

Lines 59: the authors state “The suggested “2+1” highway can use instead of such roads in order to manage traffic flow…” Yet no suggested was provided prior to the statement. Revise.

Lines 76: Define “NCHRP” as this is the first time it appears in the manuscript.

Lines 92 to 93: the authors state that “It is worth noting that the length of passing lane is an important feature of the “2+1” design that would affect both safety and mobility. Where does this come from? Where is the evidence? reference missing?

Line 94: Please revise to “not advisable”.

Simulation methodology

Again, the language of the manuscript needs revision.

Line 134: Please remove to “but” before almost.

Line 150: Please revise to “as it is simple to calculate”.

Line 150 to 152: The phrase “The time difference between two vehicles, assuming they follow their respective trajectories at their current speeds, before they crash is known as the TTC [36].” Should be stated immediately after TTC was first mentioned, which is at the beginning of line 149.

Simulation methodology

Again, the language of the manuscript needs revision.

Line 173: The language in the manuscript should ne uniform “0.50 seconds” or “0.50 s” but not both. Revise.

Results

The authors should improve their discussion of the results. The discussion within the manuscript is limited and requires significant reworking as it currently lacks coherence and is difficult to read. Otherwise:

Lines 240: Please remove to “so” before obvious.

Results and conclusion

The topic is interesting and important. However, the manuscript requires substantial rewriting and editing. The English language of the manuscript needs to be edited. The discussion within the manuscript is also limited, and requires significant reworking as it currently lacks coherence and is difficult to read.

Comments on the Quality of English Language

The English language of the manuscript needs to be edited. The authors should ask the help of native English speaking, proofreader, because there are some typo and linguistic mistakes that should be fixed.

Author Response

Thanks for your comments we addressed all the issues

Reviewer 2 Report

Comments and Suggestions for Authors

The paper describes safety and mobility performance comparison of two-plus one and two-lane two-way roads. The work is done completely by SUMO simulation, however the Authors have referred to one-hour video recorded at a real TLTW road, which was used to calibrate their model. The work is done soundly, and the results are clear. The conclusions are supported by the results. I have only a few remarks of minor importance:

1.      Figure 3. The legends and axis titles are hardly readable. Please enlarge the font slightly.

2.      Figures 4 and 5. The same.

3.      A two-plus-one configuration with opposite pass prohibited is much safer than conventional TLTW roads, but what about the costs. Why don’t apply simply two lanes for each direction? Please add a brief consideration on costs comparison.

Author Response

Thanks for your comments

Reviewer 3 Report

Comments and Suggestions for Authors

1. Author did not mentioned their exact contribution.

2. Study need more detail explanation .

3. Simulation parameters did jot well discuss how can author set the step need detail explanation.

4. Author need to explain what is new in your method how can method will significant.

5. Which term author method is best than existing methods 

Comments on the Quality of English Language

Nil

Author Response

Thanks for your comments

Reviewer 4 Report

Comments and Suggestions for Authors

The manuscript aims to show a simulation study on (2+1) and (TLTW) highways compariosn, focus on safety and mobility performance. They are using the parameters from the Egyptian highways to calibrate the software SUMO. 

In general the manuscript is well-writen, with some phrases without connection, the simulation study is well-design with variation in the traffic volume and speed, and several comparisons are made. I am just wondering about the size of the manuscript, mainly the size os tables (sometimes they are taken 1 and 2 complete pages) - see pages 14 to 17. I am not sure if they can be reduced, but they are showing a lot of information and they are not easy to read.

Some minor review:

-page2, line 57. Create a new section for "Roadway Characteristics", because this subject no longer belong to the introduction. Or just delete "Roadway Characteristics" and continues the text, since at line 110, the authors are sumarize the objectives of the manuscript.

-page2, line 72 and in the entire manuscript. The reference "Harwood D. et al. [5]" should be just [5] to keep the citation standard.

-page 7, lines 200, 202, 204 and 205. please correct the symbol "ρ" in "... the value of "ρ" is higher"... there is a weird symbol there.

Comments on the Quality of English Language

-page 4, line 162. Please consider to change the order of this prhase, because it is truncated: "Within the VISSIM simulation model, the most popular model in the literature is the Wiedemann-99 car-following model (e.g., [15,17,19,46–50], etc.), was used in this analysis."

Author Response

Thanks for your comments
